The EMBO Journal (2013) 32, 2099–2112
www.embojournal.org

# Structure of the human Parkin ligase domain in an autoinhibited state

## Tobias Wauer and David Komander*

Division of Protein and Nucleic Acid Chemistry, Medical Research Council Laboratory of Molecular Biology, Cambridge, UK

**Mutations in the protein Parkin are associated with Parkinson's disease (PD), the second most common neurodegenerative disease in men. Parkin is an E3 ubiquitin (Ub) ligase of the structurally uncharacterized RING-in-between-RING(IBR)-RING (RBR) family, which, in an HECT-like fashion, forms a catalytic thioester intermediate with Ub. We here report the crystal structure of human Parkin spanning the Unique Parkin domain (UPD, also annotated as RING0) and RBR domains, revealing a tightly packed structure with unanticipated domain interfaces. The UPD adopts a novel elongated Zn-binding fold, while RING2 resembles an IBR domain. Two key interactions keep Parkin in an autoinhibited conformation. A linker that connects the IBR with the RING2 over a 50-Å distance blocks the conserved E2 ∼ Ub binding site of RING1. RING2 forms a hydrophobic interface with the UPD, burying the catalytic Cys431, which is part of a conserved catalytic triad. Opening of intradomain interfaces activates Parkin, and enables Ub-based suicide probes to modify Cys431. The structure further reveals a putative phospho-peptide docking site in the UPD, and explains many PD-causing mutations.**

*The EMBO Journal* (2013) **32,** 2099–2112. doi:10.1038/emboj.2013.125; Published online 31 May 2013

*Subject Categories:* proteins; molecular biology of disease; structural biology

*Keywords*: E3 ligase; neurodegenerative disease; Parkin; ubiquitin; X-ray crystallography

## Introduction

Parkinson's disease (PD) is a neurodegenerative disorder characterized by loss of dopaminergic neurons from the *substantia nigra*, and appearance of α-synuclein aggregates known as Lewy bodies (Goedert *et al*, 2013). PD occurs sporadically in 1–2% of people above 65 years of age, but can also arise earlier, most commonly due to genetic predisposition of individuals (Corti *et al*, 2011). This familial form of autosomal recessive juvenile Parkinsonism (AR-JP) results from mutations in the protein kinase PINK1 (Valente *et al*, 2004), the adaptor DJ-1 (Bonifati *et al*, 2003), or, in about 50% of cases, in the E3 ubiquitin (Ub) ligase Parkin (Kitada *et al*, 1998). Heterozygous Parkin mutations have been

identified in patients with sporadic PD (Sun *et al*, 2006; Wang *et al*, 2008; reviewed in Corti *et al*, 2011).

Protein ubiquitination is an important post-translational modification that regulates most aspects of cell biology (Komander and Rape, 2012). Attachment of Ub or more frequently of polyUb chains to proteins regulates turnover, localization, complex assembly or activity of the substrate. Ubiquitination is facilitated by a three-step enzymatic cascade involving E1 Ub activating, E2 Ub conjugating enzymes, and E3 Ub ligases (Hershko and Ciechanover, 1998). Two mechanistically distinct classes of E3 ligases have been described. RING E3 ligases facilitate transfer of Ub directly from the E2 catalytic Cys to the substrate, while in HECT E3 ligases, an intermediate thioester with the E3 ligase is formed prior to substrate modification (Dye and Schulman, 2007). Recently, a hybrid mechanism involving RING-mediated formation of a ligase thioester was reported for RING-in-between-RING(IBR)-RING (RBR) E3 ligases (Wenzel *et al*, 2011). In RBR E3 ligases, a RING domain (RING1) mediates transfer of Ub from the E2 to a catalytic Cys of the RING2 domain, from which Ub is subsequently transferred to a substrate (Wenzel and Klevit, 2012). RBR E3 ligases are found in all eukaryotes, and 13 RBR E3 ligases exist in humans (Eisenhaber *et al*, 2007; Marín, 2009; Wenzel and Klevit, 2012). Prominent members of this family are HOIP and HOIL-1 that form the linear Ub chain assembly complex (LUBAC) (Kirisako *et al*, 2006), Ariadne-1 that is important for neuronal differentiation in *Drosophila* (Aguilera *et al*, 2000) and Parkin.

Genetic and cell biological work in the last decade have uncovered essential roles of Parkin and PINK1 in mitochondrial quality control (Corti *et al*, 2011; Narendra *et al*, 2012). PINK1 senses damaged mitochondria, and recruits and activates Parkin to ubiquitinate mitochondrial outer membrane proteins including mitofusins (Poole *et al*, 2010; Tanaka *et al*, 2010; Ziviani *et al*, 2010; Chen and Dorn, 2013; Sarraf *et al*, 2013) and Miro (Wang *et al*, 2011). This leads to the selective autophagy of damaged mitochondria termed mitophagy. Much evidence suggests that defects in this pathway may cause PD (Youle and Narendra, 2011). Parkin also impacts other cellular pathways, including TNFα signalling (Müller-Rischart *et al*, 2013) and Wnt/β-catenin signalling (Rawal *et al*, 2009), and Parkin was also found to be a tumour suppressor (Poulogiannis *et al*, 2010; Veeriah *et al*, 2010).

With these important biological roles, it is not surprising that Parkin E3 ligase activity is under tight regulation (Walden and Martinez-Torres, 2012), and it is clear from many studies that Parkin requires activation (Xiong *et al*, 2009; Geisler *et al*, 2010; Matsuda *et al*, 2010; Narendra *et al*, 2010; Vives-Bauza *et al*, 2010; Chaugule *et al*, 2011; Chew *et al*, 2011). Parkin regulation is in part facilitated by its domain structure (Figure 1A). A C-terminal RBR domain comprises an E2-binding RING1 domain, an IBR domain and a RING2 domain that contains the catalytic Cys residue (Cys431) for E2-mediated Ub charging. N-terminal to the RBR

*Corresponding author. Protein and Nucleic Acid Chemistry, MRC Laboratory of Molecular Biology, Francis Crick Avenue, Cambridge Biomedical Campus, Cambridge, Cambridgeshire CB2 0QH, UK. Tel.: +44 (0)1223 267160; E-mail: dk@mrc-lmb.cam.ac.uk

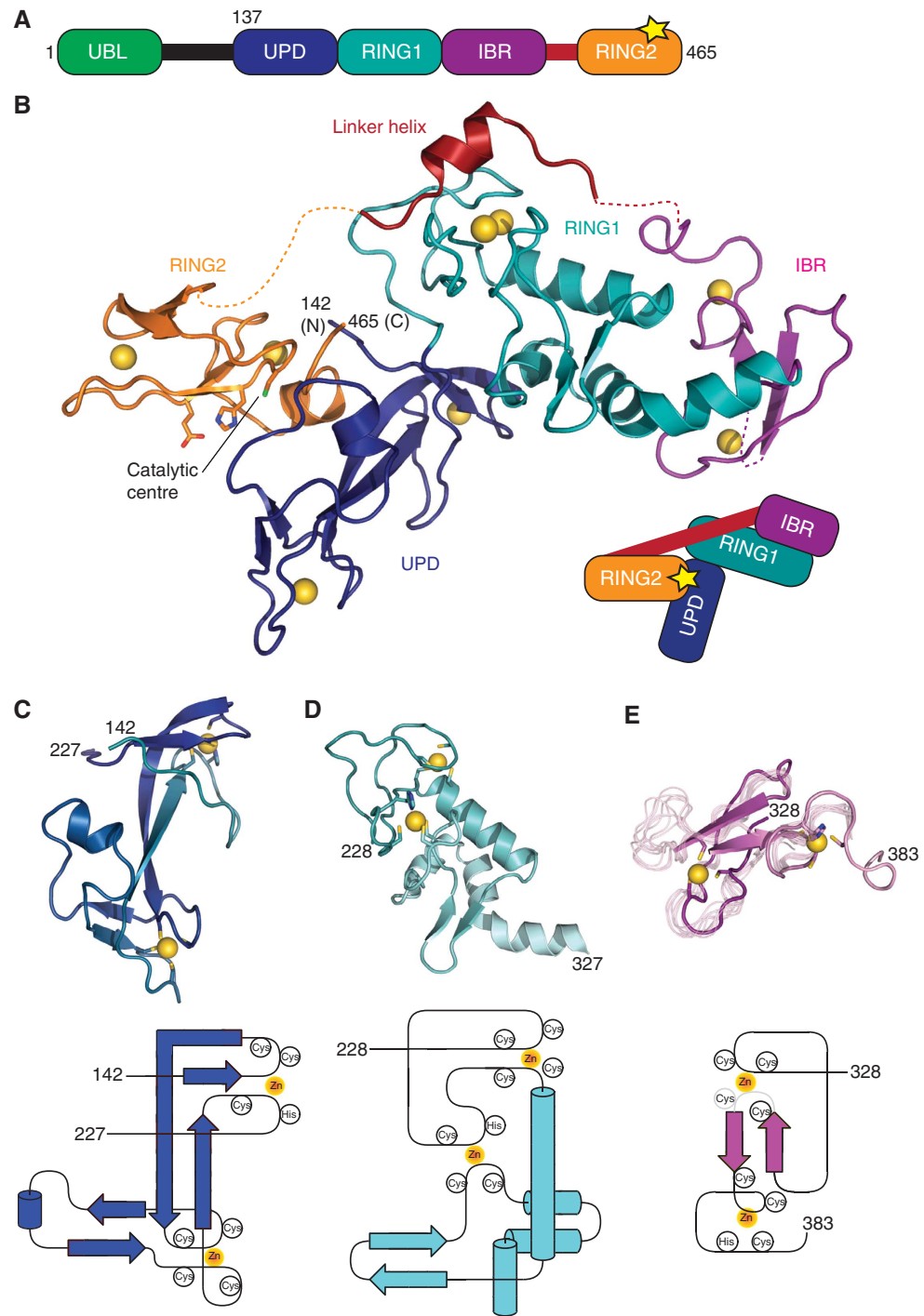

**Figure 1** Structure of Parkin. (**A**) Domain structure of Parkin. A yellow asterisk (*) indicates the catalytic Cys431 in RING2. (**B**) Structure of the Parkin UPD-RBR domain. Individual domains are coloured blue (UPD), cyan (RING1), purple (IBR) and orange (RING2). The linker helix is shown in red and the Zn atoms as yellow spheres. Putative catalytic residues in RING2 are labelled. Dotted lines indicate disordered stretches. Terminal residue numbers are indicated. A cartoon based on (**A**) depicts domain interactions. (**C**) Structure and topology of the UPD. Residues coordinating Zn atoms are shown. (**D**) Structure and topology of the extended RING1. (**E**) Structure of the crystallized IBR domain with eight superposed models from the previously described NMR ensemble (pdb-id 2jmo, Beasley *et al*, 2007). The unresolved loop in Parkin IBR is most flexible also in NMR analysis. The topology is shown below, the region in grey is disordered in the structure.

is a further Zn-binding fold termed Unique Parkin domain (UPD) or RING0 (Hristova *et al*, 2009) that interacts with PINK1 (Xiong *et al*, 2009). Finally, a Ub-like (Ubl) domain at the very N-terminus binds the RBR in *cis*, and inhibits Parkin activity (Chaugule *et al*, 2011). PINK1 was reported to phosphorylate the Ubl of Parkin to release this inhibition (Kondapalli *et al*, 2012). However, the role of phosphorylation in PINK1/Parkin function is more complicated, as PINK1 does not always phosphorylate Parkin (Vives-Bauza *et al*, 2010), PINK1 requires autophosphorylation to recruit Parkin (Okatsu *et al*, 2012), and PINK1 phosphorylates substrates to be recognized by Parkin (Chen and Dorn, 2013).

The contribution of individual Parkin domains in ubiquitination activity is however still unclear since a construct spanning IBR and RING2 but lacking the E2-interacting RING1 domain shows high *in vitro* activity, which depends on Cys431 in RING2 (Chew *et al*, 2011). While charging of the RING2 with Ub could be shown for HHARI (Wenzel *et al*, 2011) and HOIP (Smit *et al*, 2012; Stieglitz *et al*, 2012), the thioester reaction intermediate could not be observed in an *in vitro* reaction for the Parkin RBR (Wenzel *et al*, 2011). Recent data generated in an elegant cell-free system recapitulating activated PINK1 on depolarized mitochondria, showed that catalytic activity of PINK1 can lead to formation of charged Parkin in an E2-dependent manner (Lazarou *et al*, 2013). Surprisingly, Parkin C431S, which forms a stabilised Ub-loaded Parkin intermediate in this system, is not localized to mitochondria, but self-associates into an oligomeric complex in the cytosol (Lazarou *et al*, 2013).

Loss-of-function mutations in Parkin are implicated in AR-JP, and missense mutations have been an important and intensely used resource to gain insight into Parkin function (Sriram *et al*, 2005; Wang *et al*, 2005; Hampe *et al*, 2006; Matsuda *et al*, 2006, 2010; Wong *et al*, 2007; Schlehe *et al*, 2008). Many mutations affect Parkin stability but a surprising number of soluble variants appear to retain ligase activity (Sriram *et al*, 2005; Hampe *et al*, 2006; Matsuda *et al*, 2006). Molecular insights into Parkin structure would elucidate the mechanism of Parkin and of RBR E3 ligases, and could explain the connection between Parkin and AR-JP. However, while mechanisms of HECT and RING E3 ligases have been revealed in atomic detail (Dye and Schulman, 2007; Lima and Schulman, 2012), the RBR family of E3 ligases has resisted structural analysis.

We here report the structure of the catalytic fragment of Parkin spanning its UPD and RBR domains. A surprising domain disposition reveals unanticipated interfaces between UPD and RING2, which bury the catalytic Cys431 at the interface. Moreover, the IBR and RING2 domains are separated by an extended linker that occludes the canonical E2-binding site in RING1. Our structure and biochemical studies suggest that significant conformational changes are required for Parkin activation and shed light on the dysfunction of many AR-JP-associated mutations of Parkin.

## Results and discussion

### *Crystallization and structure of Parkin UPD-RBR*
RBR domains are challenging to express in bacteria due to their multiple Zn-binding folds. We were inspired by recent work on HOIP in which inclusion of a C-terminal Zn-binding region improved protein stability (Smit *et al*, 2012; Stieglitz *et al*, 2012). Parkin does not contain a C-terminal Zn-binding domain, but an N-terminal UPD Zn-binding fold that was predicted to resemble RING domains (Hristova *et al*, 2009; Figure 1A). Hristova *et al* further use limited proteolysis to identify a stable Parkin fragment corresponding to amino acids (aa) 145 to the C-terminal residue 465. We generated sufficient yields of soluble protein from human Parkin construct spanning aa 137–465, which was codon-optimized for expression in *E. coli*. Purified protein was crystallized and the structure was determined to 2.25 Å resolution using the anomalous signal of the eight coordinated zinc atoms

**Table I** Data collection, phasing and refinement statistics

| | Parkin native | Parkin Zn peak |
|---|---|---|
| *Data collection* | | |
| Space group | *H*32 | *H*32 |
| *Cell dimensions* | | |
| *a, b, c* (Å) | 168.42, 168.42, 97.18 | 168.79, 168.79, 97.42 |
| α, β, γ (deg) | 90, 90, 120 | 90, 90, 120 |
| | | *Zn peak* |
| Wavelength | 1.00000 | 1.28310 |
| Resolution (Å) | 58.33–2.25 (2.37–2.25) | 84.39–3.50 (3.69–3.50) |
| $R_{sym}$ or $R_{merge}$ | 0.089 (0.642) | 0.227 (0.560) |
| $I/\sigma I$ | 10.3 (2.3) | 27.5 (16.7) |
| Completeness (%) | 98.0 (99.1) | 100 (100) |
| Redundancy | 5.0 (5.0) | 47.8 (48.7) |
| | | |
| *Refinement* | | |
| Resolution (Å) | 48.6–2.25 | |
| No. of reflections | 24 502/1246 | |
| $R_{work}/R_{free}$ | 0.190/0.218 | |
| *No. of atoms* | | |
| Protein | 2282 | |
| Ligand/ion | 29 | |
| Water | 87 | |
| *B factors* | | |
| Protein | 46.1 | |
| Water | 39.3 | |
| *R.m.s. deviations* | | |
| Bond lengths (Å) | 0.006 | |
| Bond angles (deg) | 0.970 | |

from a single anomalous dispersion experiment (Table I; Supplementary Figure 1).

The structure of human Parkin UPD-RBR revealed four domains with unanticipated inter-domain interactions (Figure 1B). Central to the structure is the UPD (aa 142–227), a previously undescribed elongated Zn-binding fold in which two central anti-parallel β-strands arrange two Zn-coordinating loops. That the UPD is a Zn-binding fold was known, and annotation of this domain as RING0 was based on modelling it as a cross-brace structure (Hristova *et al*, 2009). We found that the UPD does not show a cross-brace topology. The two Zn atoms in the UPD are coordinated by the first and fourth, and second and third zinc-binding loops (Figure 1C). Since the UPD does not resemble a RING, and since there are no similar structures in the protein data bank, or similar sequences in the human genome, we refer to this domain as the UPD as in earlier work (Hampe *et al*, 2006).

The UPD serves as an interacting platform for the subsequent RING1 domain that binds to one tip of the UPD, and also for the RING2 domain that forms an extensive hydrophobic interface with the side of the UPD (see below).

The RING1 domain of Parkin (aa 228–327) is a variation of the canonical RING fold (Figure 1D). It contains two Zn-binding sites in cross-brace topology, and all features required for E2 binding (see below). In addition, it contains an internal solvent exposed β-hairpin insertion located after the RING domain helix (see below). Three C-terminal helices constitute a platform for IBR interaction (Figure 1B and D). The IBR domain (aa 328–383) is least well defined in the electron density, most likely due to high domain mobility. While the position of the Zn atoms is clear from the electron density, one Zn-coordinating Cys residue in the first Zn-binding site is not resolved (Figure 1E). This is consistent with NMR analysis of this domain (Beasley *et al*, 2007), which matches the observed electron density (backbone RMSD 2.0 Å) throughout the structured regions, but suggests high flexibility in this Zn-binding loop (Figure 1E). The NMR analysis indicated an extended unstructured

N-terminus (Beasley *et al*, 2007), which in the crystal structure corresponds to the RING1 helices that provide the binding site for the IBR (Figure 1B).

Intriguingly, the RING2 domain is located ∼49 Å away from the IBR domain, on the other side of the molecule (Figure 1B). A flexible linker connects these domains, and reaches across RING1, where aa 391–403 forms a short helix (see below). The intermittent residues (aa 383–390 and 404–412) span the gaps over the IBR–RING1 and RING1–RING2 interfaces, respectively, and are disordered. The disordered loop connecting the linker helix and RING2 (aa 404–412) has the potential to reach two RING2 molecules in neighbouring asymmetric units of the crystal (Supplementary Figure 1C). While in one conformation, the RING2 forms a tight hydrophobic interface with the UPD, the second conformation of RING2 would loosely attach to RING1 and IBR domains, forming few polar contacts. Mutations in the UPD or in RING2 to affect the hydrophobic interface render Parkin significantly less soluble, and hence conformation 1 was refined as the asymmetric unit.

### Insights into Parkin mechanism from the RING2 structure

The C-terminal RING2 domain shows a fold that does not resemble RING domains but rather the Parkin IBR domain (Figure 2A). Two β-sheets of two antiparallel strands each coordinate two Zn atoms, not in a cross-brace, but in a linear fashion. While the first Zn atom is coordinated by two β-hairpin loops, the second Zn is coordinated by a sequence of three Cys and one His residue, forming a 'Zn-knuckle' also observed in the IBR domain, with the differences that in RING2, the last two Zn-coordinating residues are part of a short helix (aa 455–461). The similarity to the IBR fold was confirmed by DALI analysis, which listed the HOIP IBR domain (pdb-id 2ct7 (unpublished), DALI score 2.8, backbone RMSD 0.7 Å) and the Parkin IBR NMR structure (pdb-id 2jmo, (Beasley *et al*, 2007), DALI score 1.8, backbone RMSD with crystal structure 0.4 Å) as the most similar structures (Figure 2B).

The most important feature of this domain is the catalytic Cys residue, Cys431, which is charged with Ub in RBR-mediated ubiquitination (Lazarou *et al*, 2013). Within the structure of RING2, Cys431 is in close proximity to His433 which itself interacted with Glu444, resembling a catalytic triad. All three residues are evolutionarily conserved (Supplementary Figure 2A), surface exposed and although His433 is not in hydrogen bonding distance with Cys431, only a small rotation would allow it generate a Cys431 thiolate. Moreover, analysis of conserved surface residues on the RING2 surface indicated a shallow hydrophobic groove extending from Cys431 (Figure 2C). Whether this is involved in binding to the Ub C-terminus requires further structural analysis.

Parkin terminates with a C-terminal Trp462-Phe-Asp-Val465 motif, which is the centrepiece of an extensive (686 Å$^2$) hydrophobic interface with the UPD (Figure 2D). In particular, Phe463 is buried in a hydrophobic pocket on the UPD formed by Phe146, Pro180, Trp183, Phe208 and Phe210. In addition to these hydrophobic contacts, eight hydrogen bonds between backbone and side-chain residues are formed. Ser145 of the UPD contacts His461 in the Zn-coordinating helix and the backbone of Phe463 in RING2, Tyr143 contacts Asp464, and Lys161 contacts Asp460 (Figure 2D).

Importantly, Cys431 is also located at the interface, buried from solvent, and inaccessible for transthiolation. Backbone hydrogen bonds between UPD residue Ser181 and Gly430/Cys431 contribute to a tight interface of the Cys431 containing loop (Figure 2D). Hence, the structure suggests that Cys431 cannot be charged with Ub in this conformation. This would be consistent with data by Chew *et al*, (2011) who reported that a Parkin RBR fragment (aa 237–465) was significantly more active as compared to a Parkin fragment including the UPD, however, the fragment used (aa 152–465) would lack a central β-sheet and its stability could have been compromised.

### Probing the Parkin catalytic triad

The identified putative catalytic triad in Parkin required further analysis, as similar catalytic triads have not been observed in, for example, HECT E3 ligases. GST-tagged Parkin with UPD (aa 137–465) showed weak autoubiquitination *in vitro* with E1 and UBE2L3 (Figure 2E), consistent with an autoinhibition of the RING2 domain. In contrast, a UPD-lacking RBR construct (aa 216–465) was readily modified, and most of the input Parkin protein shifted to higher molecular weight bands. This confirmed that the UPD was indeed autoinhibitory. Moreover, mutation of the putative catalytic Cys (C431A) rendered the RBR construct inactive, and Ala mutations of the catalytic triad residues His433 and Glu444 significantly reduced autoubiquitination, suggesting that the Cys431-mediated Ub transfer is compromised.

We set out to test the reactivity of the catalytic Cys residue of Parkin in a more direct manner. The catalytic triad in Parkin RING2 resembles that of Cys-based deubiquitinases (DUBs), in which a polarized catalytic His lowers the *pKa* of the catalytic Cys (Komander *et al*, 2009). In DUBs, this has been exploited by the development of Ub-based suicide inhibitors, in which the C-terminal Gly76 of Ub has been replaced with an electrophilic group such as a vinyl-sulphone (VS) or vinylmethyl ester (VME), that covalently modify the active site Cys of DUBs (Borodovsky *et al*, 2002). Ub-based suicide inhibitors have been useful reagents to identify and structurally characterize DUBs, but so far were considered as DUB specific.

Due to the presence of a similar catalytic triad, we tested whether Ub-based suicide probes would modify Parkin, and found that the crystallized fragment of Parkin did not react with Ub-based suicide inhibitors (Supplementary Figure 2B), while the same Parkin fragment including an N-terminal GST tag reacted weakly with Ub-VS (Figure 2F). Importantly, a Parkin RBR fragment lacking the UPD was significantly more reactive with Ub-VS compared to the UPD-RBR fragment (Figure 2F). This modification was dependent on Cys431, His433 and Glu444 (Figure 2F), suggesting that His433 and Glu444 indeed modulate the reactivity of Cys431 and constitute a catalytic triad.

### Conservation of a RING2 catalytic triad in other RBR E3 ligases

We wondered whether a Cys-His-Glu catalytic triad was a unique feature of Parkin, or whether it was conserved amongst other RBR enzymes. A sequence alignment of RING2 domains from all human RBR domains shows high diversity at the sequence level (Figure 3A) as noted previously (Wenzel and Klevit, 2012). The first Zn-binding site

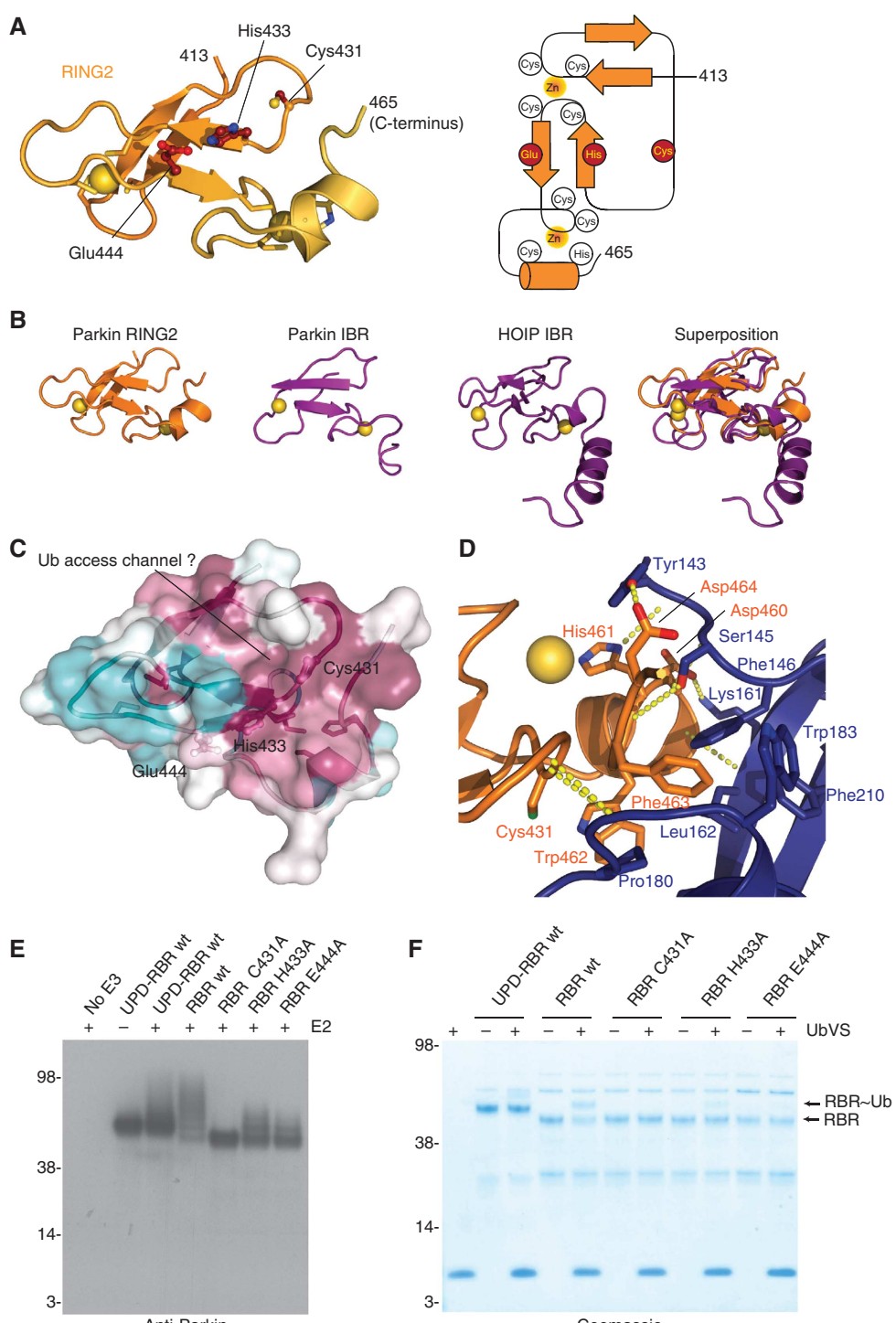

**Figure 2** Insights into RBR mechanism from RING2 structure. (**A**) RING2 structure and topology. The RING2 fold is most similar to the Parkin IBR domain (compare Figure 1E). (**B**) Comparison of RING2 (left) to the most similar structures in the pdb, the IBR domain of Parkin itself (pdb-id 2jmo, Beasley *et al*, 2007, second from left and Figure 1E) and the IBR domain of HOIP (pdb-id 2ct7, unpublished, third from left). The right image shows a superposition of all structures. (**C**) Sequence conservation in Parkin based on Supplementary Figure 2, mapped onto the surface of RING2 (purple—conserved, cyan—not conserved), suggests a potential Ub access channel. (**D**) Parkin RING2 interface with UPD. Key residues are shown and dotted lines indicate hydrogen bonds. (**E**) Parkin activity assay using GST-tagged Parkin 137–465 (UPD-RBR) or 216–465 (RBR) in the presence of E1, Ub, MgATP and with or without the E2 UBE2L3 as indicated. Western blotting for anti-Parkin reveals that while the UPD-containing fragment does not autoubiquitinate strongly, most of the Parkin RBR fragment shifts to higher molecular weight bands, and unmodified Parkin disappears. This is dependent on the three residues of the catalytic triad, mutation of which renders Parkin inactive (C431A) or less active (H433A, E444A). (**F**) Modification of Parkin variants (as in **E**) with the Ub-based suicide probe Ub-vinyl-sulphone (UbVS). A Coomassie-stained SDS–PAGE gel is shown, and bands for Parkin RBR and Ub-modified RBR (RBR∼Ub) are indicated.

is clearly conserved, and due to the linear arrangement of Zn-coordinating motifs, this also preserves the position of the catalytic Cys, which is located between the first and second Zn-binding motifs. A further downstream Cys-Xaa-Xaa-Cys motif is conserved in 12 out of 13 RBRs, but a final Zn-binding motif is not obvious. It is unclear

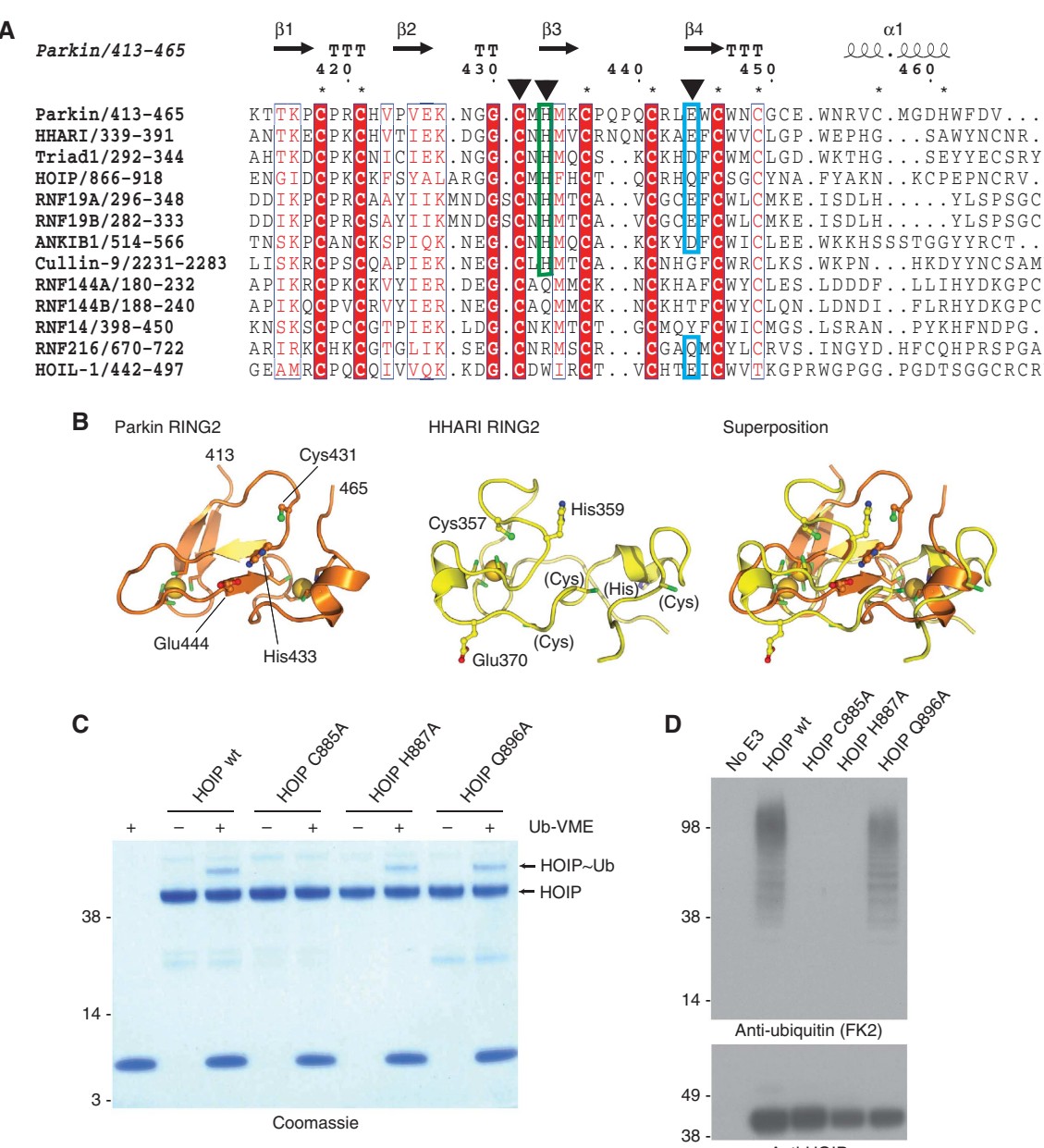

**Figure 3** RING2 domain comparison. (**A**) Sequence alignment of RING2 domains from all human RBR enzymes. Numbering and secondary structure according to Parkin. The position of Zn-binding residues is indicated by an * above the sequence, and catalytic Cys as well as putative catalytic His and Glu are indicated by arrows. (**B**) Comparison of Parkin RING2 (left) with HHARI RING2 (pdb-id 1wd2, Capili et al, 2004, middle). The right image shows a superposition. (**C**) Reactivity of HOIP RBR-LDD (aa 699–1072, Smit et al, 2012) with UbVME. A Coomassie-stained SDS–PAGE gel is shown, and HOIP as well as the modified HOIP-Ub complex are indicated. (**D**) Activity assay for HOIP mutants. Western blotting for anti-Ub (FK2, top) and for HOIP (bottom) show that HOIP assembled free Ub chains and did not autoubiquitinate. While ubiquitination activity depends on His887, probe reactivity was independent of this catalytic residue.

whether other RING2 domains also bind two Zn atoms, and the NMR structure of the HHARI RING2 domain (Capili et al, 2004) differs significantly from Parkin RING2 (backbone RMSD 7.6 Å, Figure 3B). HHARI RING2 binds only one Zn atom, despite a downstream sequence containing three un-paired Cys and one His, resembling Parkin (Figure 3A and B). Also, Cys357 in HHARI is not exposed, and it is unclear whether it can be charged with Ub in this conformation (Figure 3B).

Interestingly, 8 out of 13 RBR E3 ligases, including HHARI and HOIP, contain a His two residues downstream of the catalytic Cys, and 6 of these have a negatively charged residue

at the position corresponding to Parkin Glu444 (Figure 3A). Remarkably, the highly catalytically active minimal RBR-LDD region of HOIP (Smit et al, 2012; Stieglitz et al, 2012) was modified by Ub-VME suggesting the presence of a reactive Cys (Figure 3C), and HOIP was no longer modified when the catalytic Cys885 was mutated to Ala. In contrast to Parkin, mutation of HOIP His887 or Gln896 to Ala did not alter its ability to be modified by Ub-VME, indicating that these residues were not essential to increase the reactivity of Cys885 with suicide probes. However, activity assays with HOIP mutants revealed that HOIP C885A and H887A were inactive, while Q896A had no effect. This suggests that while His887 is not

essential for the reactivity of HOIP towards Ub-based suicide probes, this residue is important for HOIP Ub transfer. For HHARI, a stable extended RBR domain construct (aa 183–557) that was inactive in ubiquitination reactions was not modified by Ub-based suicide probes (data not shown).

Hence, Ub-based suicide inhibitors can be used to modify some RBR E3 ligases that contain a reactive, low-pKa Cys residue. This finding may benefit future studies of this E3 ligase family. However, RBR E3 ligases have not been identified to be among the targets of Ub-based inhibitors when used in cell lysates (Borodovsky, 2002) suggesting that RBR E3 ligases are perhaps kept in an inactive form in cells. Consistent with this notion, HOIP (Smit *et al*, 2012), and, as we show here, Parkin are both autoinhibited.

### The autoinhibited RING1 domain of Parkin

The model of RBR function involves a Ub-loading step that is facilitated by RING1-mediated transfer of Ub from an activated E2 to the RING2 domain (Wenzel and Klevit, 2012). The molecular requirements for RING–E2 interactions have by now been studied for several E3–E2 combinations, such as

the c-cbl–UBE2L3 interaction (Zheng *et al*, 2000). It was however only last year that the first glimpses of RING domains bound to Ub-loaded E2 were reported (Dou *et al*, 2012; Plechanovova *et al*, 2012; Pruneda *et al*, 2012). These studies revealed how a RING domain restricts movement of Ub when attached to the E2, thus activating the E2 and facilitating aminolysis.

Comparison of the RING1 domain of Parkin with RING domains of c-cbl (RMSD 2.3 Å, Zheng *et al*, 2000) or BIRC7 (RMSD 1.9 Å, Dou *et al*, 2012) confirmed its canonical features (Figure 4A). The E2 binding site in the first Zn-binding loop, and the second Zn-binding site that interacts with the Ile36-patch of Ub are conserved.

However, it became immediately apparent that in our structure of the Parkin UPD-RBR, RING1 would not be able to bind a charged E2. Superposition of Parkin with RING-E2 complex structures (Figure 4B–D) revealed significant steric clashes of E2 with the Parkin linker helix that connects IBR and RING2 (red in Figure 4B and E), as this binds RING1 across its E2 and Ub binding sites. The linker helix inserts tightly into the groove between Zn-binding sites on the

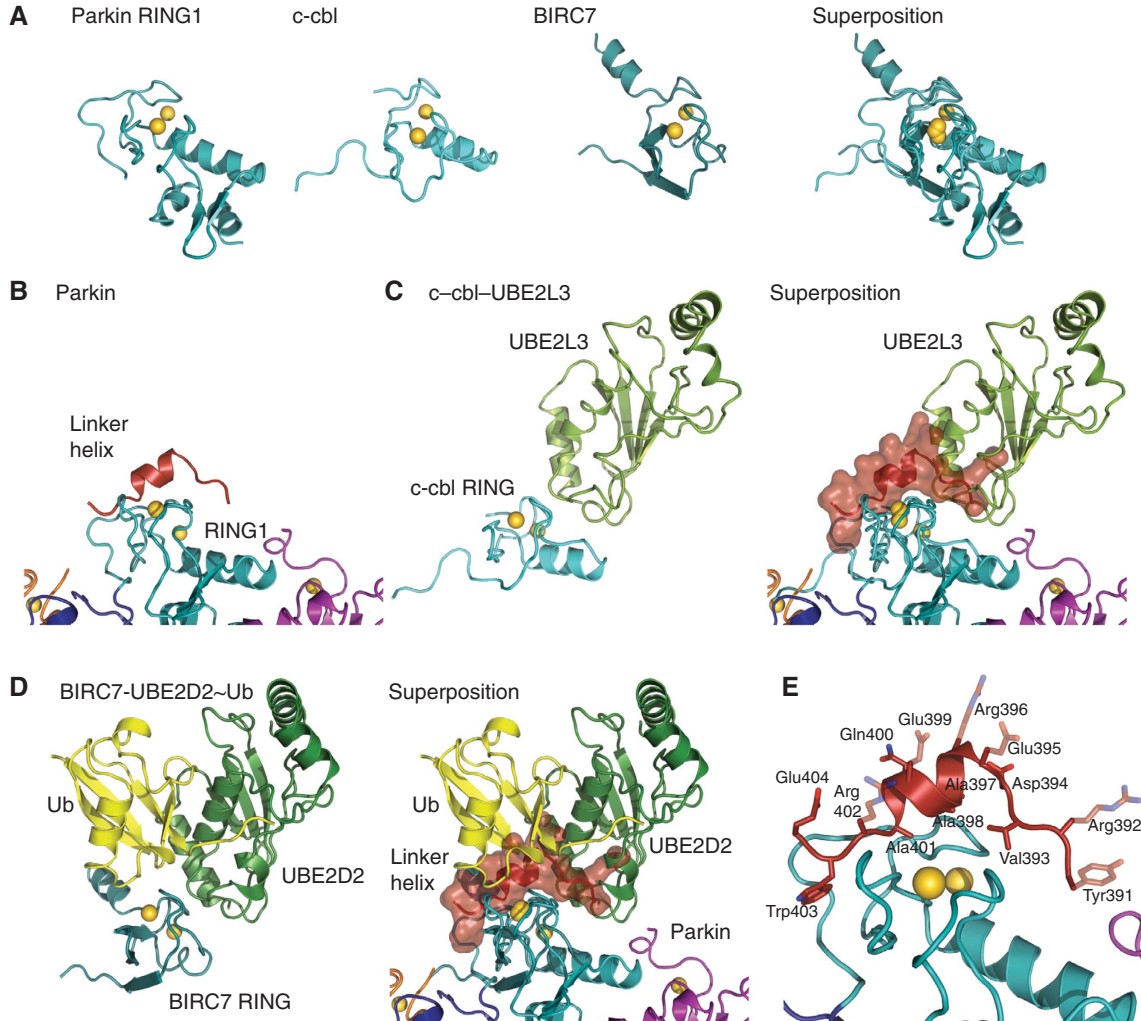

**Figure 4** RING1–E2 interactions. (**A**) Comparison of Parkin RING1 with RING domains of c-cbl (pdb-id 1fbv, Zheng *et al*, 2000) and BIRC7 (pdb-id 4auq, Dou *et al*, 2012). Zn atoms are shown as yellow spheres. (**B**) Parkin RING1 in context of the RBR, with linker helix bound. (**C**) *Left*—structure of c-cbl RING domain bound to UBE2L3, *right*—superposition of Parkin RING1 and c-cbl RING. The linker helix is shown under a semitransparent surface. (**D**) Representation as in (**C**) for BIRC7 bound to UBE2D2∼Ub. The Parkin linker helix blocks E2 and Ub interactions. (**E**) Detail of linker helix:RING1 interactions. Transparent side chains were disordered in the crystal structure and were included in their favoured rotamer to indicate the amphipathic nature of the helix.

RING1 domain, facilitated by Ala residues on the linker helix (Ala397, Ala398, Ala401) as well as Val393 and Trp403 just N- and C-terminal to the helix (Figure 4E). These hydrophobic residues interact with apolar groups on RING1, including Ile236, Ile239, Val250, Val258 and Ala291. In contrast, the solvent exposed side of the helix contains almost exclusively large, charged residues (Arg392, Glu395, Arg396, Glu399, Arg402, Glu404) (Figure 4C–E). The location and charged nature of the new surface created by linker helix binding abrogates the ability of RING1 to form canonical interactions with a Ub-charged E2. Residues N-terminal to the linker helix (aa 391–396) would obstruct crucial E2 contacts with residues in the first Zn-binding loop of RING1 (Figure 4C and D). The linker helix itself blocks the RING1 surface that engages with the Ile36 patch of Ub (Figure 4D). Hence, our structure suggests that the linker helix between IBR and RING2 inhibits E2 and E2~Ub interaction with RING1. This is consistent with biochemical findings in which we could not observe an interaction of Parkin UPD-RBR with Ub-charged UBE2L3 on analytical gel filtration, while other RBR domains (HHARI, RNF144) formed a complex with E2~Ub complex under identical conditions (data not shown). It is also interesting to note that Wenzel et al (2011) could not observe E2-dependent charging of a Parkin RBR construct. In the light of our structure, this suggests that even in absence of the UPD, the linker helix may still impair RING1–E2 interactions.

### The IBR domain and E2-independent ubiquitination

An overall view of an E2~Ub bound Parkin model (Figures 5A and 4D) revealed further interesting features. The model may explain a role of the IBR in enhancing E2 binding that has been suggested previously for Parkin (Zhang et al, 2000; Tsai et al, 2003) and other RBR E3 ligases (Moynihan et al, 1999). When the E2 is bound to Parkin RING1, the IBR is juxtaposed and could potentially enhance the binding surface with the E2 (Figure 5A).

Interestingly, especially in the light of the observed auto-inhibited RING1 domain (see above), RBR domains have also been reported to perform E2-independent ubiquitination in vitro. A Parkin fragment containing IBR, linker and RING2 domains is able to facilitate E2-independent ubiquitination in vitro (albeit at high E1 concentration), and this requires Cys431 (Chew et al, 2011). Moreover, the ubiquitination activity of the IBR-RING fragment is identical to that of an RBR fragment, and independent of E2 in the reaction, again suggesting that in an RBR fragment, RING1 may still be inhibited by the linker helix, and E2~Ub may bind by other means. Also HOIP was reported to catalyse chain formation without E2 in vitro (albeit less efficiently and at increased E1 concentration) (Smit et al, 2012).

A potentially important finding in this respect could be that both Parkin and HOIP also bind Ub non-covalently, via the IBR-RING2 linker in Parkin (Chaugule et al, 2011), and the LDD extension in HOIP (Smit et al, 2012), respectively. It is possible that this Ub-binding ability could recruit a Ub-charged E1 in a Ub-dependent manner, as has been suggested for Ub binding domain (UBD)-driven, E3-independent monoubiquitination reactions (Hoeller et al, 2007). With the reactive catalytic Cys in RING2, this may lead to transthiolation and subsequent (auto)ubiquitination in a RING1-E2 independent manner in vitro.

### Large-scale conformational changes are required for Parkin activity

The E2~Ub bound model (Figure 5A) provided further evidence that our Parkin structure represents an autoinhibited conformation. The active site Cys residue of the E2 is separated by 54 Å from Cys431 in RING2, and the E2 active site points towards the IBR rather than the RING2 (Figure 5A).

Together, the structure suggests that in order to become active, two autoinhibitory interactions need to be released in the Parkin UPD-RBR fragment. The hydrophobic RING2–UPD

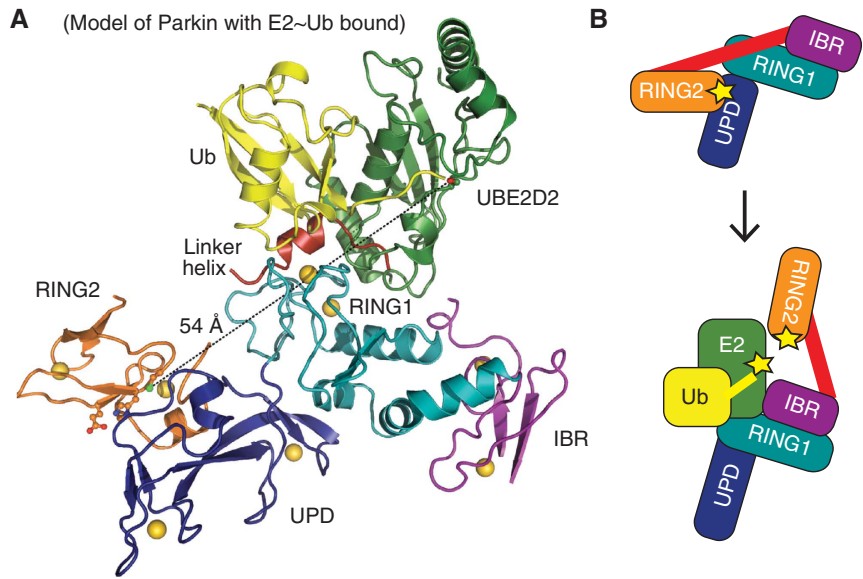

**Figure 5** Model of Parkin E2~Ub interaction. (**A**) Structure of Parkin with modelled UBE2D2~Ub from BIRC7 structure (pdb-id 4auq, Dou et al, 2012, compare Figure 3D). A dotted line connects active site residues in E2 (Cys85) and RING2 (Cys431), which are 54 Å apart. The E2 active site points towards the IBR domain. (**B**) Proposed model of Parkin activation, which involves relieving inhibitory interactions at RING1 and RING2. Yellow stars (asterisk) indicate active site Cys residues.

interface needs to open to enable access to Cys431, and the linker helix must be released from RING1 to enable E2∼Ub binding. This suggests that Parkin activation requires 'opening' of interdomain contacts within the UPD-RBR itself (Figure 5B).

### Parkin mutations give insight into Parkin mechanism

Parkin is mutated in ∼50% of patients suffering from AR-JP, as well as in a subgroup of patients that display sporadic forms of PD. A large number of mutations generate deletions, truncations and exon duplications, all of which lead to a loss of function of Parkin (Corti *et al*, 2011; Walden and Martinez-Torres, 2012). More interesting for studies on ligase function, however, are missense mutations. We collated the information available for Parkin mutations from the Leiden Open Variation Database (LOVD; http://grenada.lumc.nl/LOVD2/TPI/home.php?select_db=PARK2), the Parkinson Disease mutation database (PDmutDB; (http://www.molgen.ua.ac.be/PDmutDB/default.cfm?MT=0&ML=0&Page=Home) and (Corti *et al*, 2011), and mapped mutations situated in the crystallized construct onto our structure. Fifty-seven annotated point mutations lie within our crystallized construct (Supplementary Table I; Figure 6A).

We grouped the reported mutations by predicted effects on Parkin folding/stability, catalytic mechanism, interface formation, and mutations for which the effect cannot be easily predicted from our structure. As expected, most Parkin mutations were predicted to affect protein folding and stability, such as in Zn-coordinating residues, which are usually crucial for correct folding of Zn-binding domains. Eleven of the thirty-two Zn-coordinating residues are found mutated in patient samples (Figure 6B). In two cases (R256C, R334C), mutation introduces additional Cys or His residues next to a Zn-coordinating residue, and these likely compete for Zn binding and disrupt the fold. In addition, amino-acid changes in the core of individual domains could be predicted to disrupt domain folding, likely destabilizing Parkin (Figure 6B). Several of the Zn-coordinating and core mutants have been characterized *in vitro*, and all showed decreased solubility (Gu *et al*, 2003; Sriram *et al*, 2005; Wang *et al*, 2005; Hampe *et al*, 2006) (Supplementary Table I).

Mutations that could lead to mechanistic insights into Parkin function are those that affect the catalytic mechanism. In RING1, T240M/T240R in the first Zn-binding loop and D243N alter the conserved E2 interface, and consistently, Parkin T240R does not interact with UBE2L3 (Shimura *et al*, 2000). Some mutations in RING2 affect the catalytic triad (C431F and intriguingly also E444Q). Interestingly, Parkin C431F is less soluble, heavily ubiquitinated, but unable to ubiquitinate substrates (Sriram *et al*, 2005). This was interpreted as Parkin autoubiquitination, although it may be

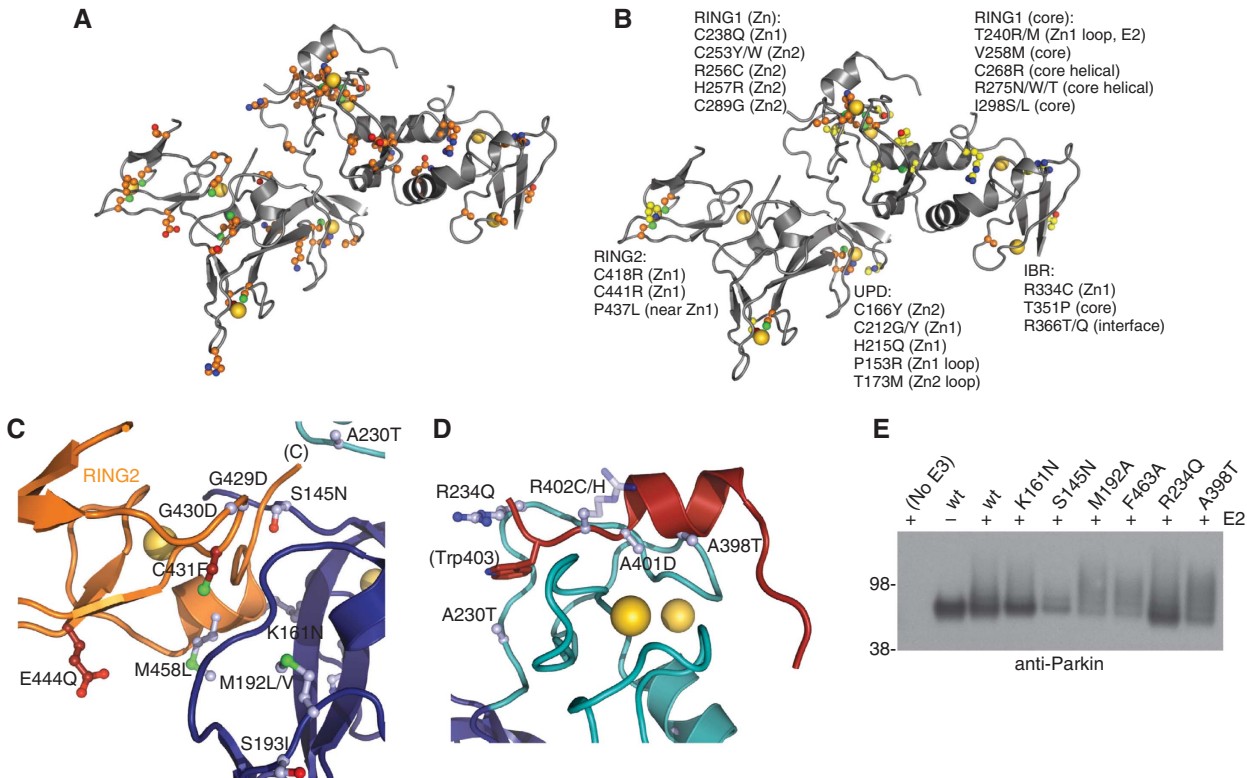

**Figure 6** Mapping PD mutations onto Parkin structure. (**A**) Parkin is shown in grey with yellow Zn atoms. Residues affected by point mutations are shown in ball-and-stick representation with orange carbon atoms. (**B**) As in (**A**), but only mutations of Zn-coordinating residues (orange) and core residues (yellow) predicted to affect folding of individual domains are shown and labelled. (**C**) Mutations of catalytic site residues (red carbon atoms) and of residues involved in the UPD: RING2 interface (blue carbon atoms). A230T mutation of RING1 may create an additional RING1: RING2 interface. (**D**) Residues in the linker helix and RING1-linker helix binding site. R402C or R402H may create an extra Zn-coordinating residue that competes for binding to RING1 Zn2. See Supplementary Figure 3 for further mutations. (**E**) Activity assay for Parkin mutants in the presence of E1, Ub, MgATP and with or without the E2 UBE2L3 as indicated (compare Figure 2E). Mutants M192A, F463A (at RING2: UPD interface) and A398T (in linker helix) were more active in autoubiquitination as compared to wt Parkin UPD-RBR. Most mutants are impaired in solubility (perhaps indicating Parkin 'opening') and have to be expressed with a GST tag to achieve soluble protein. For protein input, compare Supplementary Figure 3.

possible that other E3 ligases in mammalian cells modify the solubility-impaired protein.

### Parkin mutations affect intrinsic domain interfaces

Of particular interest are mutations in the various domain interfaces, since we speculate that opening of the interfaces might regulate Parkin activity (Figure 5B). Indeed, two hot-spots of mutations are apparent, one in the UPD–RING2 domain interface (Figure 6C), and another in the linker helix interface with the RING1 domain (Figure 6D).

As discussed above, Ser145 is a central residue at the UPD–RING2 interface, and mutation to Asn would likely alter the interface, as would mutation of Met458 to Leu. Similarly, the two Gly residues upstream of the catalytic Cys431 are found mutated to Asp potentially disrupting the interface with the UPD. Two further interesting mutations are S193I and M192L/V. These UPD residues are not directly at the interface, but stabilize the UPD loop that interacts with the Gly429-Gly430-Cys431 loop (Figure 6C).

Even more striking are mutations in the linker helix–RING1 interface (Figure 6D). Two fully conserved Ala residues, Ala398 and Ala401, are mutated in patients and the structure suggests that larger residues cannot be accommodated.

It is intriguing that several patient mutations are found at the UPD–RING2, and in the linker helix-RING1 interfaces. Such mutations could stabilize the contact locking Parkin into an autoinhibited state, or disrupt the interface, leading to 'opening' of the structure, which we predict to be required for activation (Figure 5B). This could result in activation-independent autoubiquitination, and Parkin turnover.

To test this, several Parkin mutants were generated in the UPD-RBR fragment (Figure 6E; Supplementary Figure 3A). It was immediately apparent that most mutants were impaired in their solubility when the GST tag was cleaved, suggesting that Parkin mutants indeed expose hydrophobic regions as would be predicted for 'open' domain interfaces. Accordingly, activity assays were performed with GST-tagged Parkin, and while some mutants (K161N, R234Q) showed similar autoubiquitination to wild-type Parkin, interface mutants M192A and F463A on either side of the UPD–RING2

interface, and linker helix mutation A398T showed increased autoubiquitination activity (Figure 6E), which in cells may lead to increased turnover.

In addition to the above, several not so easy to explain mutations reside in the helical extension of the RING1 domain that may affect the interface with the IBR domain (Supplementary Figure 3B). A hotspot of three mutations in the unique β-hairpin extension of RING1 is fully solvent exposed (Supplementary Figure 3B) and so is His200 at the tip of the UPD (Supplementary Figure 3C). It is unclear how these mutations disrupt Parkin function, but perhaps they affect formation of Parkin complexes.

### A putative phospho-peptide binding site in the UPD

Two patient mutations in the UPD, K161N and K211N, have been studied extensively, and were found to be soluble, expressed as well as wild-type Parkin (Sriram *et al*, 2005; Hampe *et al*, 2006) and active in autoubiquitination both *in vitro* and in cells (Hampe *et al*, 2006; Chew *et al*, 2011). However, these mutants are disabled in complex formation (Van Humbeeck *et al*, 2008), and do not localize to mitochondria (Matsuda *et al*, 2010), suggesting that they are functionally impaired.

We noticed that these residues are adjacent to each other on the backside of the UPD, solvent exposed (Figure 7A; Supplementary Figure 3B) and K161N had no impact on autoubiquitination activity (Figure 6E). Interestingly, together with nearby Arg163, they constitute a highly positively charged surface area that forms a pocket occupied by a sulphate ion from the crystallization condition (Figure 7A and B). There are several examples in which sulphates in crystal structures can be indicative of phosphate binding sites in proteins (e.g., Biondi *et al*, 2002). It is hence a possibility that Lys161, Arg163 and Lys211 are involved in binding to phosphorylated proteins.

This is interesting due to the role of the PINK1 kinase in Parkin recruitment and regulation. PINK1 binds to the Parkin UPD (Xiong *et al*, 2009) and requires autophosphorylation at two sites to recruit Parkin to mitochondria (Okatsu *et al*, 2012). Moreover, PINK1 mediated phosphorylation targets

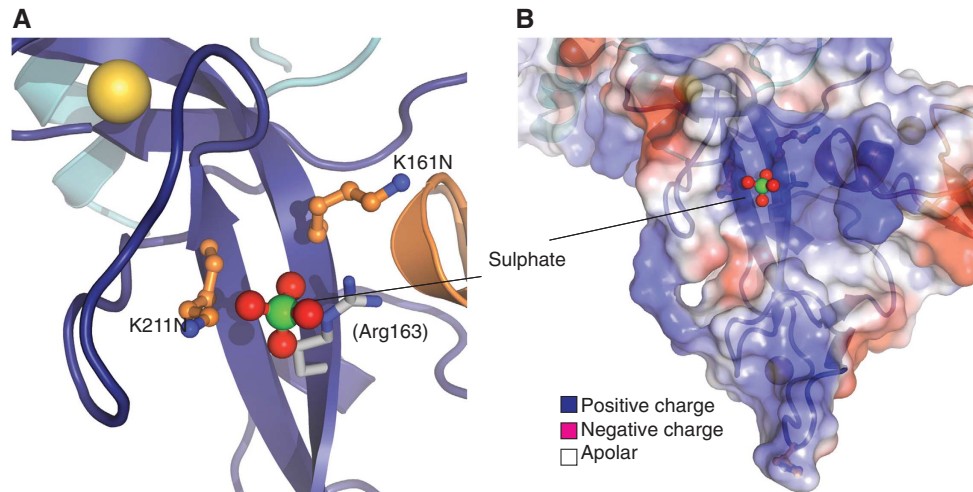

**Figure 7** A putative phospho-peptide binding site in the UPD. (**A**) Structural detail of the backside of the UPD (in relation to Figure 1B, also see Supplementary Figure 3B), showing residues Lys161 and Lys211 that are mutated in PD (orange) and Arg163 (in grey). These residues coordinate a sulphate ion from the crystallization mother liquor (shown in ball-and-stick, with green sulphur and red oxygen). (**B**) Electrostatic surface potential of the UPD (generated with PyMol) showing the positively charged pocket in the UPD that is occupied by a sulphate ion.

proteins to Parkin (Chen & Dorn 2013), and this could be mediated by the putative phosphopeptide binding site. It will be interesting to study whether Lys161/Lys211 are involved in binding phosphorylated proteins.

### Multi-step activation of Parkin

Based on our structure, we describe a two-fold autoinhibition of Parkin, where the linker helix blocks E2 access to RING1, and the UPD blocks access to Cys431 in RING2. It appears that for full Parkin activation, both interactions have to be released, and that the structure of Parkin has to open (Figure 5B).

Walden and colleagues recently revealed a further mechanism of autoinhibition, in which the N-terminal Ubl of Parkin interacts in *cis* with a region corresponding to the IBR-RING2 linker (Chaugule *et al*, 2011). Interestingly, the reported region that binds the Ubl includes the linker helix, which we here show binds and inhibits RING1. However, the solvent-accessible side of the linker helix contains numerous charged residues (Figure 4E), making this an unlikely interaction surface.

In the light of our structure, this still does not explain why a construct without the Ubl shows activity (Chaugule *et al*, 2011), which is in fact contradictory to several previous reports showing that a Parkin fragment lacking the Ubl is inactive (Corti *et al*, 2003; Cha *et al*, 2005; Henn *et al*, 2007). Moreover, we show here that at least autoubiquitination activity is comparably weak (Figures 2E and 6E), even in the presence of a GST tag that often acts as a Ub acceptor in autoubiquitination assays (Yang *et al*, 2005). *In vitro* GST-RING mediated autoubiquitination reactions or chain assembly by HOIP (Figure 3D) are significantly more processive.

The Ub ligase activity of Parkin has been intensely studied; however, the results from such studies are hard to interpret due to the various autoinhibitory mechanisms, and differences in performing the assays. Indeed, it is often not clear whether the (auto)ubiquitination observed is due to RING1-mediated discharging of an E2 onto the catalytic Cys431, or directly onto another nearby Lys or Cys residue. This can be partly addressed by using UBE2L3, which discharges exclusively onto Cys (Wenzel *et al*, 2011), and such autoubiquitination is Cys431 dependent (Figure 2E). However, many experiments in the literature have been performed using UBE2D (UbcH5), and since UBE2D discharges onto both Cys and Lys residues, autoubiquitination and perceived activity could be Cys431 independent.

As mentioned above, it is also not clear whether RING1 is involved in the discharge of the E2, as RING1-independent ubiquitination has been demonstrated (Chew *et al*, 2011). The mentioned Ubl-binding region mapped by Chaugule *et al* (2011) also binds Ub, and this might directly attract a Ub-charged E2 enzyme to perform RING1-independent ubiquitination (see above).

Hence, it seems essential to understand whether Parkin-mediated (auto)ubiquitination is RING1 and RING2 dependent. Our structural findings in combination with cell-free Parkin activation assays (Lazarou *et al*, 2013) should allow separation of the reported Parkin activities mediated by RING1, IBR and RING2.

## Conclusions

The E3 ligase Parkin is under intense investigation due to its involvement in PD and important roles in mitochondrial maintenance (Corti *et al*, 2011; Narendra *et al*, 2012). The here reported structure of a Parkin fragment comprising UPD and RBR domains explains many biochemical findings, but also reveals several unexpected features, and highlights the extensive regulation of Parkin, since several mechanisms of autoinhibition seem to be in place to keep Parkin inactive.

It is clear that our structure of Parkin has to be devoid of catalytic activity, yet the fact that our tagged Parkin fragments (which lack Ubl-mediated autoinhibition) show (weak) catalytic activity, and the notion that large conformational changes are required for activity, indicates a much more dynamic interplay between Parkin domains. It is clear that our structure is just a first step to understanding Parkin activation, and future work should focus on generating active protein, and activated structures of Parkin.

A large fraction of the loss-of-function Parkin mutations found in PD patients destabilize the protein by affecting folding of the complicated Zn-bound architecture. More interesting are those mutations for which loss-of-function cannot be immediately attributed to decreased Parkin stability, and we find several of such instances, most notably in a to-be-confirmed docking site for phospho-peptides in the UPD. An interesting case is provided further by mutations that would affect inter-domain interactions. Our model of Parkin activation and biochemical data indicate that domain-opening mutations might be 'gain-of-function' with respect to ligase activity. Since Parkin autoubiquitinates, this may lead to increased turnover, and lower steady state protein levels. Another explanation would be that aberrant opening of the Parkin interfaces, that is, in the absence of an activating binding partner or bound substrate, would simply expose hydrophobic surfaces that lead to Parkin aggregation. Indeed, our preliminary experiments suggest that 'open' Parkin is highly insoluble when expressed in bacteria.

The key question that remains is how Parkin is activated, for instance by PINK1 once it arrives at mitochondria, as Parkin activators must release a variety of safety belts that keep Parkin inactive. Moreover, the active form of Parkin may be oligomeric, as recently reported (Van Humbeeck *et al*, 2008; Lazarou *et al*, 2013), and we would expect this to be distinct from the autoinhibited form reported here. Understanding oligomeric status and composition of the complexes containing active Parkin should help to further understand the cellular roles of Parkin in mitophagy, and its pathophysiological roles in neurodegenerative disease.

## Materials and methods

### Molecular biology

cDNA for full-length human Parkin was obtained by gene synthesis (Genscript) and codon-optimized for expression in *E. coli*. The Parkin UPD-RBR, aa 137–465 was cloned into pOPINK (Berrow *et al*, 2007), which contains an N-terminal 3C-protease-cleavable GST tag, using the Infusion HD Kit (Clontech). The following primers were used: Parkin 137–465 fwd 5′-AAGTTCTGTTTCAGGG CCCGGCTGGTCGTTCGATCTACAACAGCTTCTATGTGTAC-3′ and Parkin 137–465 rev 5′-ATGGTCTAGAAAGCTTTACACATCAAACCAG TGGTCACCCATACACACAC-3′. The RBR and the C-terminal region of HOIP (residues 699–1072) (Smit *et al*, 2012; Stieglitz *et al*, 2012) were cloned into pOPINK with the Infusion HD Kit (Clontech). HOIP and Parkin mutants were generated in pOPINK by site-directed mutagenesis using the Quikchange method with KOD HotStart DNA polymerase according to the manufacturer's protocol. All constructs were confirmed by DNA sequencing.

### Protein expression and purification

Parkin UPD-RBR was expressed in Rosetta2 pLacI cells, which were grown to an $OD_{600}$ of 1.0 at 37°C. In all cases 200 μM zinc chloride was added upon induction with 50 μM IPTG, and cells were grown at 16°C for 12 h. The cells were lysed by sonication in 270 mM sucrose, 10 mM glycerol 2-phosphate disodium, 50 mM NaF, 14 mM β-mercaptoethanol, 50 mM Tris (pH 8.0). Lysate was cleared by centrifugation (45 000 g, 30 min, 4°C), applied to Glutathione Sepharose 4B beads (GE Healthcare, 0.5 ml/l of culture), and incubated under agitation for 1 h at 4°C. Beads were washed with high salt buffer (500 mM NaCl, 10 mM DTT, 25 mM Tris (pH 8.5)) and buffer A (200 mM NaCl, 10 mM DTT, 25 mM Tris (pH 8.5)). GST-tagged Parkin constructs used for biochemical analysis was eluted by incubating the beads with buffer A containing 30 mM glutathione for 0.5 h at 4°C. Untagged Parkin used for crystallization was cleaved on the beads by incubating with GST-3C protease for 12 h at 4°C. After diluting the buffer to 75 mM NaCl, 10 mM DTT, 25 mM Tris (pH 8.5), the protein was applied to anion exchange (RESOURCE Q, GE Life Sciences) and eluted with a linear gradient of 75–600 mM NaCl in 10 mM DTT, 25 mM Tris (pH 8.5). Protein containing fractions were pooled, concentrated and applied to gel filtration (Superdex 75, GE Life Sciences) in Buffer A. The protein eluted as a single peak at the size expected for a monomer, and was concentrated using a 3-kDa MWCO spin concentrator (VWR) and flash frozen in liquid nitrogen.

HOIP 699-1072 was expressed and purified as described for Parkin by cleaving the protein from Glutathione Sepharose 4B beads and applying the eluted protein to gel filtration.

### Crystallization

Human Parkin UPD-RBR was crystallized at a concentration of ∼4 mg/ml using vapour diffusion. Initial screening in nano-litre volume gave two hits in similar conditions that were optimized to obtain diffraction quality crystals. Final crystals were grown from mixing 0.5 μl protein with 0.5 μl mother liquor (1.6 M lithium sulphate, 10 mM magnesium chloride, 50 mM MES (pH 5.4)). Prior to vitrification in a nitrogen cryostream, crystals were briefly soaked in mother liquor containing 15% glycerol.

### Data collection and structure determination

Diffraction data were collected the Diamond Light Source (Harwell, UK), beamline I-04. Crystals belonged to space group *H32* containing one molecule per asymmetric unit, and diffracted to a maximum resolution of 2.25 Å (at 100 K, λ = 1.0000 Å). For phasing, a highly redundant data set to 3.5 Å was collected at the Zn peak wavelength (at 100 K, λ = 1.28310 Å), which allowed phasing by single anomalous dispersion due to the eight bound Zn atoms, using the SHELX package (Zheng *et al*, 2000; Rodríguez *et al*, 2009). Automated model building in ArpWarp (Langer *et al*, 2008; Dou *et al*, 2012) was followed by rounds of manual building in coot (Emsley *et al*, 2010; Dou *et al*, 2012) and refinement in Phenix (Adams *et al*, 2011). Final statistics are shown in Table I. There are no Ramachandran outliers in the structure.

### Modification with Ub-based suicide probes

Ubiquitin suicide probes UbC2Cl, UbC3Cl and UbC3Br were generated as described previously (Borodovsky *et al*, 2002; Akutsu *et al*, 2011). In brief, 200 μl Ub-thioester was mixed with 40 mg 2-bromoethylamine hydrochloride solved in 200 μl phosphate-buffer saline (PBS, pH 4.8) and the reaction was initiated by adding 80 μl 2 M NaOH. After 15 min on ice, the probes were dialysed against 200 mM NaCl, 25 mM Tris pH 8.5 using Slide-A-Lyzer Dialysis Cassettes (Thermo Scientific). Ub-VS and Ub-VME were obtained from Boston Biochem, dissolved in 0.08% TFA.

For time-course experiments in Supplementary Figure 2B, 10 μl of protein (0.4 mg/ml) in buffer A was mixed with 4 μl probe (0.83 mg/ml). The reaction was incubated at room temperature and stopped at the indicated time points by adding 4× LDS sample buffer (Invitrogen). In all, 7 μl of sample was resolved on NuPAGE 4–12% Bis-Tris gels (Invitrogen).

The reaction of HOIP mutants with UbVME was performed by mixing 0.28 mg/ml of protein with 0.2 mg/ml of the probe in reaction buffer (50 mM Tris (pH 8.5), 200 mM NaCl, 10 mM DTT). The reaction took place for 1 h at 37°C and was quenched by adding 4× LDS sample buffer (Invitrogen). In all, 7 μl of sample was loaded on NuPAGE 4–12% Bis-Tris gels (Invitrogen).

Parkin mutants were reacted with UbVS by mixing 0.14 mg/ml protein with 0.1 mg/ml probe, in Parkin reaction buffer (50 mM Tris (pH 7.4), 200 mM NaCl, 10 mM DTT). The assay was performed at 37°C and quenched after 30 min by adding 4× LDS sample buffer (Invitrogen). In all, 4 μl of sample was applied on NuPAGE 4–12% Bis-Tris gels (Invitrogen). All gels were stained with Instant Blue SafeStain (Expedeon).

### Ubiquitination assay

Ubiquitination assays were performed by mixing 100 nM E1, 6 μM UBE2L3 (UBCH7), 4 μM GST-Parkin or HOIP, 0.25 mg/ml Ubiquitin and 10 mM ATP in ubiquitination buffer (40 mM Tris pH 7.5, 10 mM $MgCl_2$, 0.6 mM DTT). The reaction took place for 1 h at 37°C and was quenched by adding 4× LDS sample buffer (Invitrogen). The sample was further diluted 1:30 and resolved on NuPAGE 4–12% Bis-Tris gels (Invitrogen). After western blotting, membranes were incubated with anti-Parkin (anti-PRK8, from Abcam), anti-Ubiquitin (FK2, from Millipore) or anti-HOIP (Sigma), and analysed by enhanced chemiluminescence (ECL Prime, Roche).

### Accession numbers

Coordinates and structure factors have been deposited with the protein data bank, accession code 4bm9.

### Note added in proof

Our data is consistent with recent structural work on rat Parkin, which additionally resolves the position of the Ubl domain (Trempe *et al*, 2013).

### Supplementary data

Supplementary data are available at *The EMBO Journal* Online (http://www.embojournal.org).

## Acknowledgements

We are grateful to Francesco Melandri and Boston Biochem for providing Ub-VS and Ub-VME probes. Moreover we would like to thank members of the Komander lab for reagents and helpful discussions, and Paul Elliott for help with crystallization and data collection. Crystallographic data were collected at the Diamond Light Source, beamline I-04. This work was supported by the Medical Research Council (U105192732), the European Research Council (309756), the Lister Institute for Preventive Medicine, and the EMBO Young Investigator Programme.

*Author contributions*: TW and DK designed and performed experiments, analysed the data and wrote the manuscript.

## Conflict of interest

The authors declare that they have no conflict of interest.

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
