## [Review Process File · The EMBO Journal]

Manuscript EMBO-2013-85110

Structure of the human Parkin ligase domain in an autoinhibited state

Tobias Wauer, David Komander

Corresponding author: David Komander, MRC Laboratory of Molecular Biology

Review timeline:

Submission date:	18 March 2013
Editorial Decision:	08 April 2013
Revision received:	06 May 2013
Accepted:	07 May 2013

Editor: Hartmut Vodermaier

Transaction Report:

1st Editorial Decision

08 April 2013

Thank you again for submitting your Parkin structure manuscript for our consideration. Copied below, I am forwarding you the reports from two expert referees who have now assessed the study. Since both referees appreciate the quality and overall importance of this work, we should in principle be happy to consider this study further for publication, provided that you adequately revise it in response to the referees' comments. As you will see, the reviewers raise a limited number of both specific and general concerns, including the absence of more functional and/or mechanistic follow-up analyses. I realize that you may well have already obtained such further data, whose inclusion will clearly strengthen paper. On the other hand, I would be less concerned about referee 1's issue with insufficient new information on the possible regulatory roles of the missing N-terminal Ubl domain, as I would consider this beyond the scope of this first description of the complete catalytic domain. I would be happy to further discuss any plans and suggestions for revising based on the referees' comments with you.

Please be reminded that it is our policy that competing manuscripts published during our regular single three months revision period will have no negative impact on our final assessment of your revised study. However, we request that you contact the editor as soon as possible upon publication of any related work, to discuss how to proceed. Should you foresee a problem in meeting this three-month deadline, please let us know in advance and we may be able to grant an extension.

Thank you for the opportunity to consider this work, and please do not hesitate to contact me in case you should have any additional question regarding this decision or the reports. I look forward to your revision.

REFeree REPORTS:

Referee #1

In their study, Wauer and Komander report on the crystal structure of N-terminally truncated parkin spanning the UPD (Unique Parkin Domain, also known as RING0) and RBR domain (aa 137 - 465). The data revealed that Parkin is kept in an autoinhibited conformation mediated by two key interactions. RING2 and the UPD form a hydrophobic interface that buries the catalytic residue C431. In addition, a linker helix between the IBR and RING2 binds to RING1, thereby blocking the E2-ub binding site.

The authors also found that the UPD does not display a cross-brace topology, as predicted earlier by Hristova et al. (*J. Biol. Chem.*, 2009), but forms an elongated Zn-binding fold with two central anti-parallel beta-strands.

The study by Wauer and Komander revealed novel and interesting features of the UPD-RBR domain which help to explain the autoinhibited state of recombinant parkin previously reported by Chaugule et al. (*EMBO J.*, 2011). Unfortunately, it does not address the controversial role of the N-terminal UBL domain in parkin folding and activity, since the authors were obviously unable to determine the structure of full-length Parkin. Chaugule et al. reported that the UBL domain of parkin is responsible for autoinhibition of parkin, based on their observation that recombinant DeltaUBL Parkin shows more auto-ubiquitination in *in vitro* ubiquitination assays. However, the DeltaUBL Parkin mutant shows no or at least a significantly impaired E3 ligase activity towards several Parkin substrates in cellular models (Corti et al., *Hum. Mol. Gen.*, 2003; Huynh et al., *Hum. Mol. Gen.*, 2003; Cha et al., *PNAS* 2005; Henn et al., *J. Neurosci.*, 2007) and no functional activity (Darios et al., *Hum. Mol. Gen.*, 2003; Trempe et al., *Mol. Cell*, 2009; Kim et al., *J. Clin. Invest.*, 2012; Mueller-Rischart et al., *Mol. Cell*, 2013). Interestingly, DeltaUBL Parkin immunoprecipitated from cells has no auto-ubiquitination activity *in vitro* (Shimura et al., *Nature Genetics*, 2000). Thus, a structural study on full-length Parkin might be helpful to explain this discrepancy.

Without doubt, insight into the structure of recombinant parkin is helpful to understand the effect of loss-of-function mutations and the regulation of parkin. A weak point of the study is that does not include any functional data (*in vivo* and *in vitro* ubiquitination assays). The authors only speculate on how pathogenic Parkin mutations may impact on Parkin structure and activation, but do not support their statements by experimental data including Parkin mutants. Moreover, several studies reported that parkin occurs in a higher molecular weight complex *in vivo* so that the structure of recombinant Parkin *in vitro* might be different from the *in vivo* situation. The authors should mention this in their Conclusions section.

Minor point:

In their introduction the author state that PINK1 phosphorylates Parkin, which has been reported in some but not all studies addressing this question (see Lazarou et al, *Dev. Cell*, 2012; Thomas et al., *Hum. Mol. Gen.*, 2011; Vives-Bauza, *PNAS*, 2010). Phosphorylation of parkin has also been shown to inhibit parkin function (Ko et al., 2010, *PNAS*; Imam, *J. Neurosci.*, 2011 and some other publications). Therefore, the authors should be more precise about this controversial issue.

Referee #2

The manuscript by Wauer and Komander entitled 'Structure of the autoinhibited Parkin catalytic domain' presents the 2.25 Å structure of a Parkin construct derived from amino acids (137-465) that includes the UPD, RING1, IBR and C-terminal RING2 but which lacks a functionally relevant N-terminal Ubl domain. The structural results present what appears to be an autoinhibited state of Parkin as both the catalytic cysteine and presumed E2~Ub binding site are occluded by other Parkin elements, a result consistent with the observation that this fragment of Parkin is not active as a ligase *in vitro*. The structure illuminates several new features and is a welcome addition to the literature. In addition it provides a number of testable hypotheses with respect to Parkin activation and the mechanism of Ub transfer from E2s and Ub transfer to targets. The paper is largely descriptive and

speculative with respect to potential mechanisms for autoinhibition and the catalytic mechanism because none of the central hypotheses were explored biochemically by probing the presumed catalytic triad and activation mechanism (perhaps the A398T and A401D mutations). The paper would be more impactful if this was done although the structure will certainly inspire others to do so if the authors decide not to generate this data.

Figure 1B should also include a label for the N-terminus.

The authors state that there is absolutely no ambiguity in assignment of the linker helix to RING2 and the IBR. While there does not appear to be any ambiguity in the connection between residues 383 to 391 inspection of the termini and lattice mates reveal two potential connections between the linker and IBR (404 to 414), one spanning $18.4 \approx$ and the other (to an adjacent lattice mate) spanning $25.2 \approx$. While I agree that the most likely connection is the one assigned by the authors, it is certainly possible that 9 amino acids could span $25.2 \approx$. This point should be discussed in more detail.

If a catalytic triad indeed exists in this ligase it would differ substantially from other ligases, some additional discussion and explicit mention of this fact is warranted.

Response to Reviewer comments

We would like to thank the reviewers for their positive evaluation of our crystal structure analysis of Parkin. We agree with their comments that our structural work would be nicely complemented by biochemical analysis, such as ubiquitination assays. In this revised version, we now provide biochemical verification for the key findings of our study.

We perform both (auto)ubiquitination assays on a variety of Parkin and HOIP variants. Moreover, we also developed an assay to confirm our discovery of a putative catalytic triad in Parkin. The catalytic triad resembles that of Cys deubiquitinases (DUBs), for which Ub-based suicide inhibitors have been developed that covalently modify the active site Cys. In these reagents, the C-terminus of Ub is exchanged for an electrophilic group (such as Ub-vinyl sulfone (Ub-VS), or a vinyl methyl ester (Ub-VME)). There are no reports known to us that these compounds target anything but DUBs.

We here show that Parkin and HOIP can be directly and covalently modified by ubiquitin-based suicide inhibitors.

Our crystallised fragment of Parkin was not modified by Ub suicide probes, as the UPD blocks the catalytic Cys431. Addition of a GST-tag resulted in slight modification of a UPD-RBR construct by Ub-VS, and additional removal of the UPD increased Parkin modification, suggesting that Cys431 was now accessible. Importantly, Ub-VS modification was dependent on the identified catalytic Cys, His and Glu residue in Parkin, confirming our discovery of a catalytic triad. The importance of the catalytic triad was independently confirmed by autoubiquitination assays, which also showed that a Parkin fragment lacking the UPD was significantly more active, consistent with our structure.

We also found that a catalytically active form of HOIP was covalently modified by Ub-VME. Mutation of the catalytic Cys885 in HOIP abrogated the modification, but while the predicted catalytic triad residues His887 and Gln896 residue did not inhibit probe reactivity, the H887A mutant of HOIP was devoid of catalytic activity.

Finally, we tested six Parkin mutations in intra-domain interfaces, and several of these mutants activate Parkin in autoubiquitination assays.

Together, our biochemical analysis confirms the key findings of our paper.

A detailed response to the Reviewer comments follows.

Referee #1

In their study, Wauer and Komander report on the crystal structure of N-terminally truncated parkin spanning the UPD (Unique Parkin Domain, also known as RING0) and RBR domain (aa 137 - 465). The data revealed that Parkin is kept in an autoinhibited conformation mediated by two key interactions. RING2 and the UPD form a hydrophobic interface that buries the catalytic residue C431. In addition, a linker helix between the IBR and RING2 binds to RING1, thereby blocking the E2-ub binding site.

The authors also found that the UPD does not display a cross-brace topology, as predicted earlier by Hristova et al. (J. Biol. Chem., 2009), but forms an elongated Zn-binding fold with two central anti-parallel beta-strands.

The study by Wauer and Komander revealed novel and interesting features of the UPD-RBR domain which help to explain the autoinhibited state of recombinant parkin previously reported by Chaugule et al. (EMBO J., 2011), Unfortunately, it does not address the controversial role of the N-terminal UBL domain in parkin folding and activity, since the authors were obviously unable to determine the structure of full-length Parkin. Chaugule et al. reported that the UBL domain of parkin is responsible for autoinhibition of parkin, based on their observation that recombinant DeltaUBL Parkin shows more auto-ubiquitination in in vitro ubiquitination assays. However, the DeltaUBL Parkin mutant shows no or at least a significantly impaired E3 ligase activity towards several Parkin substrates in cellular models (Corti et al., Hum. Mol. Gen., 2003; Huynh et al., Hum. Mol. Gen., 2003; Cha et al., PNAS 2005; Henn et al., J. Neurosci., 2007) and no functional activity (Darios et al., Hum. Mol. Gen., 2003; Trempe et al., Mol. Cell, 2009; Kim et al., J. Clin. Invest., 2012; Mueller-Rischart et al., Mol. Cell, 2013). Interestingly, DeltaUBL Parkin immunoprecipitated from cells has no auto-ubiquitination activity in vitro (Shimura et al., Nature Genetics, 2000). Thus, a structural study on full-length Parkin might be helpful to explain this discrepancy.

Without doubt, insight into the structure of recombinant parkin is helpful to understand the effect of loss-of-function mutations and the regulation of parkin.

We would like to thank this reviewer for their support, and for pointing out the many controversies in the Parkin literature.

Indeed, we feel that our data on the catalytic UPD-RBR construct very much agree with all the reports that the Reviewer mentions, as we indeed show structurally, and now functionally, that a Ubl-lacking construct is still autoinhibited.

In this revised version of the manuscript, we show that Parkin without Ubl is significantly less active as compared to Parkin without the UPD, and we show why this is in our doubly-inhibited structure.

We have now discussed this in more detail in our revised section on “Multi-step activation of Parkin”, and we refer to several mentioned references.

We have been unable to generate full-length Parkin in bacteria, and hence cannot discuss the full-length Parkin structure or the role of the Ubl domain. We fully agree that a structure of full-length Parkin would be fantastic, but we have been unable to obtain this so far. However, we felt that our structure of the extended catalytic domain of Parkin was worth reporting, as it already gave interesting insights and raised many questions that now need to be addressed.

A weak point of the study is that does not include any functional data (in vivo and in vitro ubiquitination assays). The authors only speculate on how pathogenic Parkin mutations may impact on Parkin structure and activation, but do not support their statements by experimental data including Parkin mutants.

We had based our arguments on the many reports that addressed the activity of Parkin in vitro and in cells, and tried as good as possible to reconcile the published literature. We have now been more critical with our assessment of published experiments, and in the last paragraph of the paper discuss the difficulties in defining ‘active’ Parkin.

As explained above, we spent a significant effort to test our hypotheses biochemically in this revised version of the paper. We now provide evidence that Parkin contains a catalytic triad, that this may be conserved in other RBRs, and that patient mutations in intra-domain interfaces lead to more active Parkin. We are unable to test these mutants in vivo, as we are not set up for this at this stage, but this will be an important future aspect of our work.

Moreover, several studies reported that parkin occurs in a higher molecular weight complex in vivo so that the structure of recombinant Parkin in vitro might be different from the in vivo situation. The authors should mention this in their Conclusions section.

This is indeed interesting, and especially a recent paper by Lazarou and colleagues (JCB 2013) also provides interesting data on this.

As requested, we have mentioned this in the conclusions section, where we now state :

Moreover, the active form of Parkin may be oligomeric, as recently reported (Lazarou *et al*, 2013; Van Humbeeck *et al*, 2008), and we would expect this to be distinct from the autoinhibited form reported here.

Minor point:

In their introduction the author state that PINK1 phosphorylates Parkin, which has been reported in some but not all studies addressing this question (see

Lazarou et al, Dev. Cell, 2012; Thomas et al., Hum. Mol. Gen., 2011; Vives-Bauza, PNAS, 2010). Phosphorylation of parkin has also been shown to inhibit parkin function (Ko et al., 2010, PNAS; Imam, J. Neurosci., 2011 and some other publications). Therefore, the authors should be more precise about this controversial issue.

We have refined our description of this. We were quite convinced by data presented in the recent paper by Kondapalli et al (OpenBiol 2012), but have now provided a more balanced perspective how the interplay between PINK1 and Parkin is more complicated.

Referee #2

The manuscript by Wauer and Komander entitled 'Structure of the autoinhibited Parkin catalytic domain' presents the 2.25 Å structure of a Parkin construct derived from amino acids (137-465) that includes the UPD, RING1, IBR and C-terminal RING2 but which lacks a functionally relevant N-terminal Ubl domain. The structural results present what appears to be an autoinhibited state of Parkin as both the catalytic cysteine and presumed E2~Ub binding site are occluded by other Parkin elements, a result consistent with the observation that this fragment of Parkin is not active as a ligase in vitro. The structure illuminates several new features and is a welcome addition to the literature. In addition it provides a number of testable hypotheses with respect to Parkin activation and the mechanism of Ub transfer from E2s and Ub transfer to targets. The paper is largely descriptive and speculative with respect to potential mechanisms for autoinhibition and the catalytic mechanism because none of the central hypotheses were explored biochemically by probing the presumed catalytic triad and activation mechanism (perhaps the A398T and A401D mutations). The paper would be more impactful if this was done although the structure will certainly inspire others to do so if the authors decide not to generate this data.

We would like to thank the reviewer for their positive comments on our work, and we agree that the identification of a putative catalytic triad in RBR E3 ligases is very interesting and testable.

As described above, we now show that the catalytic Cys residue of HOIP and Parkin can be modified covalently by Ub-based suicide inhibitors that target low-pKa Cys residues. Indeed, for Parkin, this depends on all three catalytic triad residues, mutation of which also greatly impairs activity. We show that the requirement for a Cys/His dyad is conserved in HOIP, as mutation of the HOIP catalytic triad renders HOIP inactive.

This is indeed a novel insight into this E3 ligase family, as Ub suicide probes do not show such reactivity with Cys-based E1 or E2 enzymes or HECT E3 ligases. Our insights also have some implication for the use of Ub-based

suicide inhibitors, although we note that RBR E3 ligases have never been reported to be amongst the targets of Ub-based suicide inhibitors in cell lysates. It is possible that RBRs are usually inhibited (as in Parkin, or as in HOIP, which interacts in *cis* with inhibitory domains (Smit et al, EMBO J 2012)), or that their reactivity in cells is lower. Ub suicide probes will also be interesting to generate structures of 'charged' RBR E3 ligases.

- Figure 1B should also include a label for the N-terminus.

We have done this.

- The authors state that there is absolutely no ambiguity in assignment of the linker helix to RING2 and the IBR. While there does not appear to be any ambiguity in the connection between residues 383 to 391 inspection of the termini and lattice mates reveal two potential connections between the linker and IBR (404 to 414), one spanning 18.4 Å and the other (to an adjacent lattice mate) spanning 25.2 Å. While I agree that the most likely connection is the one assigned by the authors, it is certainly possible that 9 amino acids could span 25.2 Å. This point should be discussed in more detail.

This was a great comment, and we appreciated this only after the paper was submitted. In early stages of the refinement we had indeed used an alternative RING2 in the model building, and we never really looked back at this after the hydrophobic UPD-RING2 interface became obvious, hence the strong statement. The Reviewer is right that the packing allows for a potential second conformation of RING2, in which it would interact with the IBR domain. We have now mentioned this and show this in a new Supplementary Figure. We wondered whether this may represent a structure that resembles a predicted 'open' conformation of Parkin (as in our Model in Figure 5B), but the second conformation of RING2 would clash with E2 binding. Our mutagenesis shows that point mutants in the UPD-RING2 interface render the protein insoluble, so we believe that the structure shown in Fig 1 is the predominant if not only conformation of Parkin purified from cells.

- If a catalytic triad indeed exists in this ligase it would differ substantially from other ligases, some additional discussion and explicit mention of this fact is warranted.

We agree, and have now discussed this at length in our new section describing the biochemical experiments.

Acceptance letter

07 May 2013

Thank you for submitting your revised manuscript for our consideration. I have now had a chance to look through it and to assess your responses to the referees' original comments in detail. I am pleased to see that you have been broadly able to answer and extend the points raised by the reviewers. We are therefore happy to accept your manuscript for publication as an article in The EMBO Journal.
